# Making Sense of Reinforcement Learning and Probabilistic Inference

**Brendan O'Donoghue**[*†]        **Ian Osband** [*†]        **Catalin Ionescu** [†]

## Abstract

Reinforcement learning (RL) combines a control problem with statistical estimation: The system dynamics are not known to the agent, but can be learned through experience. A recent line of research casts 'RL as inference' and suggests a particular framework to generalize the RL problem as probabilistic inference. Our paper surfaces a key shortcoming in that approach, and clarifies the sense in which RL can be coherently cast as an inference problem. In particular, an RL agent must consider the effects of its actions upon future rewards and observations: The exploration-exploitation tradeoff. In all but the most simple settings, the resulting inference is computationally intractable so that practical RL algorithms must resort to approximation. We demonstrate that the popular 'RL as inference' approximation can perform poorly in even very basic problems. However, we show that with a small modification the framework does yield algorithms that can provably perform well, and we show that the resulting algorithm is equivalent to the recently proposed K-learning, which we further connect with Thompson sampling.

## 1 Introduction

Probabilistic inference is a procedure of making sense of uncertain data using Bayes' rule. The optimal control problem is to take actions in a known system in order to maximize the cumulative rewards through time. Probabilistic graphical models (PGMs) offer a coherent and flexible language to specify causal relationships, for which a rich literature of learning and inference techniques have developed (Koller & Friedman, 2009). Although control dynamics might also be encoded as a PGM, the relationship between action planning and probabilistic inference is not immediately clear. For inference, it is typically enough to specify the system and pose the question, and the objectives for learning emerge automatically. In control, the system and objectives are known, but the question of how to approach a solution may remain extremely complex (Bertsekas, 2005).

Perhaps surprisingly, there is a deep sense in which inference and control can represent a dual view of the same problem. This relationship is most clearly stated in the case of linear quadratic systems, where the Ricatti equations relate the optimal control policy in terms of the system dynamics (Welch et al., 1995). In fact, this connection extends to a wide range of systems, where control tasks can be related to a dual inference problem through rewards as exponentiated probabilities in a distinct, but coupled, PGM (Todorov, 2007; 2008). A great benefit of this connection is that it can allow the tools of inference to make progress in control problems, and vice-versa. In both cases the connections provide new insights, inspire new algorithms and enrich our understanding (Toussaint & Storkey, 2006; Ziebart et al., 2008; Kappen et al., 2012).

Reinforcement learning (RL) is the problem of learning to control an unknown system (Sutton & Barto, 2018). Like the control setting, an RL agent should take actions to maximize its cumulative rewards through time. Like the inference problem, the agent is initially uncertain of the system dynamics, but can learn through the transitions it observes. This leads to a fundamental tradeoff: The agent may be able to improve its understanding through exploring poorly-understood states and actions, but it may be able to attain higher immediate reward through exploiting its existing knowledge (Kearns & Singh, 2002). In many ways, RL combines control and inference into a general framework for decision making under uncertainty. Although there has been ongoing research

---

[*]These authors contributed equally to this work.

[†]DeepMind, London, UK, {`bodonoghue`, `iosband`, `cdi`}`@google.com`

in this area for many decades, there has been a recent explosion of interest as RL techniques have made high-profile breakthroughs in grand challenges of artificial intelligence research (Mnih et al., 2013; Silver et al., 2016).

A popular line of research has sought to cast 'RL as inference', mirroring the dual relationship for control in known systems. This approach is most clearly stated in the tutorial and review of Levine (2018), and provides a key reference for research in this field. It suggests that a *generalization* of the RL problem can be cast as probabilistic inference through inference over exponentiated rewards, in a continuation of previous work in optimal control (Todorov, 2009). This perspective promises several benefits: A probabilistic perspective on rewards, the ability to apply powerful inference algorithms to solve RL problems and a natural exploration strategy. In this paper we will outline an important way in which this perspective is incomplete. This shortcoming ultimately results in algorithms that can perform poorly in even very simple decision problems. Importantly, these are not simply technical issues that show up in some edge cases, but fundamental failures of this approach that arise in even the most simple decision problems.

In this paper we revisit an alternative framing of 'RL as inference'. In fact, we show that the *original* RL problem was already an inference problem all along.[1] Importantly, this inference problem includes inference over the agent's future actions and observations. Of course, this perspective is not new, and has long been known as simply the Bayes-optimal solution, see, *e.g.*, Ghavamzadeh et al. (2015). The problem is that, due to the exponential lookahead, this inference problem is fundamentally intractable for all but the simplest problems (Gittins, 1979). For this reason, RL research focuses on computationally efficient approaches that maintain a level of statistical efficiency (Furmston & Barber, 2010; Osband et al., 2017).

We provide a review of the RL problem in Section 2, together with a simple and coherent framing of RL as probabilistic inference. In Section 3 we present three approximations to the intractable Bayes-optimal policy. We begin with the celebrated Thompson sampling algorithm, then we review the popular 'RL as inference' framing, as presented by Levine (2018), and highlight a clear and simple shortcoming in this approach. Finally, we review K-learning (O'Donoghue, 2018), which we re-interpret as a modification to the RL as inference framework that provides a principled approach to the statistical inference problem, as well as a presenting a relationship with Thompson sampling. In Section 4 we present computational studies that support our claims.

## 2  REINFORCEMENT LEARNING

We consider the problem of an agent taking actions in an unknown environment in order to maximize cumulative rewards through time. For simplicity, this paper will model the environment as a finite horizon, discrete Markov Decision Process (MDP) $M = (\mathcal{S}, \mathcal{A}, \mathcal{R}, \mathcal{P}, H, \rho)$.[2] Here $\mathcal{S} = \{1, .., S\}$ is the state space, $\mathcal{A} = \{1, .., A\}$ is the action space and each episode is of fixed length $H \in \mathbb{N}$. Each episode $\ell \in \mathbb{N}$ begins with state $s_0 \sim \rho$ then for timesteps $h = 0, .., H-1$ the agent selects action $a_h$, observes transition $s_{h+1}$ with probability $\mathcal{P}(s_{h+1}, s_h, a_h) \in [0, 1]$ and receives reward $r_{h+1} \sim \mathcal{R}(s_h, a_h)$, where we denote by $\mu(s_h, a_h) = \mathbb{E}r_{h+1}$ the mean reward. We define a policy $\pi$ to be a mapping from $\mathcal{S}$ to probability distributions over $\mathcal{A}$ and write $\Pi$ for the space of all policies. For any timestep $t = (\ell, h)$, we define $\mathcal{F}_t = (s_0^0, a_0^0, r_1^0, .., s_{h-1}^\ell, a_{h-1}^\ell, r_h^\ell)$ to be the sequence of observations made before time $t$. An RL algorithm maps histories to policies $\pi_t = \mathrm{alg}(\mathcal{S}, \mathcal{A}, \mathcal{F}_t)$.

Our goal in the design of RL algorithms is to obtain good performance (cumulative rewards) for an unknown $M \in \mathcal{M}$, where $\mathcal{M}$ is some *family* of possible environments. Note that this is a different problem from typical 'optimal control', that seeks to optimize performance for one particular known MDP $M$; although you might still fruitfully apply an RL *algorithm* to solve problems of that type. For any environment $M$ and any policy $\pi$ we can define the action-value function,

$$Q_h^{M,\pi}(s, a) = \mathbb{E}_{\pi,M} \left[ \sum_{j=h+1}^{H} r_j \mid s_h = s, a_h = a \right]. \tag{1}$$

---

[1]Note that, unlike control, connecting RL with inference will not involve a separate 'dual' problem.
[2]This choice is for clarity; continuous, infinite horizon, or partially-observed environments do not alter our narrative.

Where the expectation in (1) is taken with respect to the action selection $a_j$ for $j > h$ from the policy $\pi$ and evolution of the fixed MDP $M$. We define the value function $V_h^{M,\pi}(s) = \mathbb{E}_{\alpha \sim \pi} Q_h^{M,\pi}(s, \alpha)$ and write $Q_h^{M,\star}(s, a) = \max_{\pi \in \Pi} Q_h^{M,\pi}(s, a)$ for the optimal Q-values over policies, and the optimal value function is given by $V_h^{M,\star}(s) = \max_a Q_h^{M,\star}(s, a)$.

In order to compare algorithm performance across different environments, it is natural to normalize in terms of the *regret*, or shortfall in cumulative rewards relative to the optimal value,

$$\text{Regret}(M, \text{alg}, L) = \mathbb{E}_{M,\text{alg}} \left[ \sum_{\ell=1}^{L} \left( V_0^{M,\star}(s_0^\ell) - \sum_{h=1}^{H} r_h^\ell \right) \right]. \tag{2}$$

This quantity depends on the unknown MDP $M$, which is fixed from the start and kept the same throughout, but the expectations are taken with respect to the dynamics of $M$ and the learning algorithm alg. For any particular MDP $M$, the optimal regret of zero can be attained by the non-learning algorithm $\text{alg}_M$ that returns the optimal policy for $M$.

In order to assess the quality of a reinforcement learning algorithm, which is designed to work across some *family* of $M \in \mathcal{M}$, we need some method to condense performance over a set to a single number. There are two main approaches to this:

$$\text{BayesRegret}(\phi, \text{alg}, L) = \mathbb{E}_{M \sim \phi} \text{Regret}(M, \text{alg}, L), \tag{3}$$

$$\text{WorstCaseRegret}(\mathcal{M}, \text{alg}, L) = \max_{M \in \mathcal{M}} \text{Regret}(M, \text{alg}, L), \tag{4}$$

where $\phi$ is a prior over the family $\mathcal{M}$. These differing objectives are often framed as Bayesian (average-case) (3) and frequentist (worst-case) (4) RL [3]. Although these two settings are typically studied in isolation, it should be clear that they are intimately related through the choice of $\mathcal{M}$ and $\phi$. Our next section will investigate what it would mean to 'solve' the RL problem. Importantly, we show that both frequentist and Bayesian perspectives already amount to a problem in probabilistic inference, without the need for additional re-interpretation.

## 2.1 SOLVING THE RL PROBLEM THROUGH PROBABILISTIC INFERENCE

If you want to 'solve' the RL problem, then formally the objective is clear: find the RL algorithm that minimizes your chosen objective, (3) or (4). To anchor our discussion, we introduce a simple decision problem designed to highlight some key aspects of reinforcement learning. We will revisit this problem setting as we discuss approximations to the optimal policy.

**Problem 1** (One unknown action). *Fix $N \in \mathbb{N} \geq 3, \epsilon > 0$ and define $\mathcal{M}_{N,\epsilon} = \{M_{N,\epsilon}^+, M_{N,\epsilon}^-\}$. Both $M^+$ and $M^-$ share $\mathcal{S} = \{1\}, H = 1$ and $\mathcal{A} = \{1, .., N\}$; they only differ through their rewards:*

$$\mathcal{R}^+(1) = 1, \quad \mathcal{R}^+(2) = +2, \quad \mathcal{R}^+(a) = 1 - \epsilon \ \text{for} \ a = 3, .., N,$$
$$\mathcal{R}^-(1) = 1, \quad \mathcal{R}^-(2) = -2, \quad \mathcal{R}^-(a) = 1 - \epsilon \ \text{for} \ a = 3, .., N.$$

*Where $\mathcal{R}(a) = x \in \mathbb{R}$ is a shorthand for deterministic reward of $x$ when choosing action $a$.*

Problem 1 is extremely simple, it involves no generalization and no long-term consequences: It is an independent bandit problem with only one unknown action. For *known* $M^+, M^-$ the optimal policy is trivial: Choose $a_t = 2$ in $M^+$ and $a_t = 1$ in $M^-$ for all $t$. An RL agent faced with *unknown* $M \in \mathcal{M}$ should attempt to optimize the RL objectives (3) or (4). Unusually, and only because Problem 1 is so simple, we can actually compute the *optimal* solutions to both in terms of $L$ (the total number of episodes) and $\phi = (p^+, p^-)$ where $p^+ = \mathbb{P}(M = M^+)$, the probability of being in $M^+$.

For $L > 3$ an optimal *minimax* (minimizing the worst-case regret) RL algorithm is to first choose $a_0 = 2$ and observe $r_1$. If $r_1 = 2$ then you know you are in $M^+$ so pick $a_t = 2$ for all $t = 1, 2..$, for $\text{Regret}(L) = 0$. If $r_1 = -2$ then you know you are in $M^-$ so pick $a_t = 1$ for all $t = 1, 2..$, for $\text{Regret}(L) = 3$. The worst-case regret of this algorithm is 3, which cannot be bested by any algorithm.

---

[3]Some frequentist results are high-probability bounds on the worst case rather than true worst-case bounds, but this distinction is not important for our purposes

Actually, the same RL algorithm is also *Bayes*-optimal for any $\phi = (p^+, p^-)$ provided $p^+ L > 3$. This relationship is not a coincidence. All admissible solutions to the worst-case problem (4) are given by solutions to the average-case (3) for some 'worst-case' prior $\tilde{\phi}$ (Wald, 1950). As such, for ease of exposition, our discussion will focus on the Bayesian (or average-case) setting. However, readers should understand that the same arguments apply to the worst-case objective.

In Problem 1, the key probabilistic inference the agent must consider is the effects of it own *actions* upon the future rewards, *i.e.*, whether it has chosen action 2. Slightly more generally, where actions are independent and episode length $H = 1$, the optimal RL algorithm can be computed via Gittins indices, but these problems are very much the exception (Gittins, 1979). In problems with generalization or long-term consequences, computing the Bayes-optimal solution is computationally intractable. One example of an algorithm that converges to Bayes-optimal solution in the limit of infinite computation is given by Bayes-adaptive Monte-Carlo Planning (Guez et al., 2012). The problem is that, even for very simple problems, the lookahead tree of interactions between actions, observations and algorithmic updates grows exponentially in the search depth (Strehl et al., 2006). Worse still, direct computational approximations to the Bayes-optimal solution can fail exponentially badly should they fall short of the required computation (Munos, 2014). As a result, research in reinforcement learning amounts to trying to find computationally tractable approximations to the Bayes-optimal policy that maintain some degree of statistical efficiency.

## 3 Approximations for computational and statistical efficiency

The exponential explosion of future actions and observations means solving for the Bayes-optimal solution is computationally intractable. To counter this, most computationally efficient approaches to RL simplify the problem at time $t$ to only consider inference over the data $\mathcal{F}_t$ that has been gathered prior to time $t$. The most common family of these algorithms are 'certainty equivalent' (under an identity utility): They take a point estimate for their best guess of the environment $\hat{M}$, and try to optimize their control given these estimates $V^{\hat{M},\star}$. Typically, these algorithms are used in conjunction with some dithering scheme for random action selection (*e.g.*, epsilon-greedy), to mitigate premature and suboptimal convergence (Watkins, 1989). However, since these algorithms do not prioritize their exploration, they may take exponentially long to find the optimal policy (Osband et al., 2014).

In order for an RL algorithm to be statistically efficient, it must consider the value of information. To do this, an agent must first maintain some notion of epistemic uncertainty, so that it can direct its exploration towards states and actions that it does not understand well (O'Donoghue et al., 2018). Here again, probabilistic inference finds a natural home in RL: We should build up posterior estimates for the unknown problem parameters, and use this *distribution* to drive efficient exploration.[4]

### 3.1 Thompson sampling

One of the oldest heuristics for balancing exploration with exploitation is given by Thompson sampling, or probability matching (Thompson, 1933). Each episode, Thompson sampling (TS) randomly selects a policy according to the probability it is the optimal policy, conditioned upon the data seen prior to that episode. Thompson sampling is a simple and effective method that successfully balances exploration with exploitation (Russo et al., 2018).

Implementing Thompson sampling amounts to an inference problem at each episode. For each $s, a, h$ define the binary random variable $\mathcal{O}_h(s, a)$ where $\mathcal{O}_h(s, a) = 1$ denotes the event that action $a$ is optimal for state $s$ in timestep $h$.[5] The TS policy for episode $\ell$ is thus given by the inference problem,

$$\pi^{\text{TS}} \sim \mathbb{P}(\mathcal{O} \mid \mathcal{F}_\ell), \tag{5}$$

where $\mathbb{P}(\mathcal{O} \mid \mathcal{F}_\ell)$ is the *joint* probability over all the binary optimality variables (hereafter we shall suppress the dependence on $\mathcal{F}_\ell$). To understand how Thompson sampling guides exploration let us consider its performance in Problem 1 when implemented with a uniform prior $\phi = (\frac{1}{2}, \frac{1}{2})$. In the

---

[4]For the purposes of this paper, we will focus on *optimistic* approaches to exploration, although more sophisticated information-seeking approaches merit investigation in future work (Russo & Van Roy, 2014).

[5]For the problem definition in Section 2 there is always a deterministic optimal policy for $M$.

first timestep the agent samples $M_0 \sim \phi$. If it samples $M^+$ it will choose action $a_0 = 2$ and learn the true system dynamics, choosing the optimal arm thereafter. If it samples $M^-$ it will choose action $a_0 = 1$ and repeat the identical decision in the next timestep. Note that this procedure achieves BayesRegret 2.5 according to $\phi$, but *also* worst-case regret 3, which matches the optimal minimax performance despite its uniform prior.

Recent interest in TS was kindled by strong empirical performance in bandit tasks (Chapelle & Li, 2011). Following work has shown that this algorithm satisfies strong Bayesian regret bounds close to the known lower bounds for MDPs, under certain assumptions (Osband & Van Roy, 2017; 2016). However, although much simpler than the Bayes-optimal solution, the inference problem in (5) can still be prohibitively expensive. Table 1 describes one approach to performing the sampling required in (5) implicitly, by maintaining an explicit model over MDP parameters. This algorithm can be computationally intractable as the MDP becomes large and so attempts to scale Thompson sampling to complex systems have focused on *approximate* posterior samples via randomized value functions, but it is not yet clear under which settings these approximations should be expected to perform well (Osband et al., 2017). As we look for practical, scalable approaches to posterior inference one promising (and popular) approach is known commonly as 'RL as inference'.

Table 1: Model-based Thompson sampling.

| Before episode $\ell$ | Sample $M_\ell = (\mathcal{S}, \mathcal{A}, \mathcal{R}^\ell, \mathcal{P}^\ell, H, \rho) \sim \phi \mid \mathcal{F}_\ell$ |
|---|---|
| Bellman equation | $Q_h^\ell(s,a) = \mu^\ell(s,a) + \sum_{s'} \mathcal{P}^\ell(s',s,a)V_{h+1}^\ell(s')$ $V_h^\ell(s) = \max_a Q_h^\ell(s,a)$ |
| Policy | $\pi_h^{\text{TS}}(s,a) \in \operatorname{argmax} Q_h^\ell(s,a)$ |

## 3.2 THE 'RL AS INFERENCE' FRAMEWORK AND ITS LIMITATIONS

The computational challenges of Thompson sampling suggest an approximate algorithm that replaces (5) with a parametric distribution suitable for expedient computation. It is possible to view the algorithms of the 'RL as inference' approach in this light (Rawlik et al., 2013; Todorov, 2009; Toussaint, 2009; Deisenroth et al., 2013; Fellows et al., 2019); see Levine (2018) for a recent survey. These algorithms choose to model the probability of optimality according to,

$$\tilde{\mathbb{P}}(\mathcal{O}_h(s,a)|\tau_h(s,a)) \propto \exp\left(\sum_{(s',a')\in\tau_h(s,a)} \beta\mathbb{E}^\ell\mu(s',a')\right). \tag{6}$$

for some $\beta > 0$, where $\tau_h(s,a)$ is a trajectory (a sequence of state-action pairs) starting from $(s,a)$ at timestep $h$, and where $\mathbb{E}^\ell$ denotes the expectation under the posterior at episode $\ell$. With this potential in place one can perform Bayesian inference over the unobserved 'optimality' variables, obtaining posteriors over the policy or other variables of interest. This presentation of the RL as inference framework is slightly closer to the one in Deisenroth et al. (2013, §2.4.2.2) than to Levine (2018), but ultimately it produces the same family of algorithms. We provide such a derivation in the appendix for completeness.

Applying inference procedures to (6) leads naturally to RL algorithms with some 'soft' Bellman updates, and added entropy regularization. We describe the general structure of these algorithms in Table 2. These algorithmic connections can help reveal connections to policy gradient, actor-critic, and maximum entropy RL methods (Mnih et al., 2016; O'Donoghue et al., 2017; Haarnoja et al., 2017; 2018; Eysenbach et al., 2018). The problem is that this resultant 'posterior' derived using (6) does not generally bear any close relationship to the agent's epistemic probability that $(s,a,h)$ is optimal.

Table 2: Soft Q-learning.

| | |
|---|---|
| Bellman equation | $\tilde{Q}_h(s,a) = \mathbb{E}^\ell \mu(s,a) + \sum_{s'} \mathbb{E}^\ell \mathcal{P}(s', s, a)\tilde{V}_{h+1}(s')$ |
| | $\tilde{V}_h(s) = \beta^{-1} \log \sum_a \exp \beta \tilde{Q}_h(s,a)$ |
| Policy | $\pi_h^{\text{SQ}}(s,a) \propto \exp \beta \tilde{Q}_h(s,a)$ |

To understand how 'RL as inference' guides decision making, let us consider its performance in Problem 1. Practical implementations of 'RL as inference' estimate $\mathbb{E}^\ell \mu$ through observations. For $N$ large, and without prior guidance, the agent is then extremely unlikely to select action $a_t = 2$ and so resolve its epistemic uncertainty. Even for an informed prior $\phi = (\frac{1}{2}, \frac{1}{2})$ action selection according to the exploration strategy of Boltzmann dithering is unlikely to sample action 2 for which $\mathbb{E}^\ell \mu(2) = 0$ (Levine, 2018; Cesa-Bianchi et al., 2017). This is because the $N-1$ 'distractor' actions with $\mathbb{E}^\ell \mu \geq 1 - \epsilon$ are much more probable under the Boltzmann policy.

This problem is the same problem that afflicts most dithering approaches to exploration. 'RL as inference' as a framework does not incorporate an agents epistemic uncertainty, and so can lead to poor policies for even simple problems. While (6) allows the construction of a dual 'posterior distribution', this distribution does not generally bear any relation to the typical posterior an agent should compute conditioned upon the data it has gathered, *e.g.*, equation (5). Despite this shortcoming RL as inference has inspired many interesting and novel techniques, as well as delivered algorithms with good performance on problems where exploration is not the bottleneck (Eysenbach et al., 2018). However, due to the use of language about 'optimality' and 'posterior inference' *etc.*, it may come as a surprise to some that this framework does not truly tackle the Bayesian RL problem. Indeed, algorithms using 'RL as inference' can perform very poorly on problems where accurate uncertainty quantification is crucial to performance. We hope that this paper sheds some light on the topic.

### 3.3 Making sense of 'RL as Inference' via K-learning

In this section we suggest a subtle alteration to the 'RL as inference' framework that develops a coherent notion of optimality. The K-learning algorithm was originally introduced through a risk-seeking exponential utility (O'Donoghue, 2018). In this paper we re-derive this algorithm as a principled approximate inference procedure with clear connections to Thompson sampling, and we highlight its similarities to the 'RL as inference' framework. We believe that this may offer a road towards combining the respective strengths of Thompson sampling and the 'RL as inference' frameworks. First, consider the following approximate conditional optimality probability at $(s, a, h)$:

$$\tilde{\mathbb{P}}(\mathcal{O}_h(s,a)|Q_h^{M,\star}(s,a)) \propto \exp \beta Q_h^{M,\star}(s,a), \tag{7}$$

for some $\beta > 0$, and note that this is conditioned on the random variable $Q_h^{M,\star}(s,a)$. We can marginalize over possible Q-values yielding

$$\tilde{\mathbb{P}}(\mathcal{O}_h(s,a)) = \int \tilde{\mathbb{P}}(\mathcal{O}_h(s,a)|Q_h^{M,\star}(s,a))d\mathbb{P}(Q_h^{M,\star}(s,a)) \propto \exp G_h^Q(s,a,\beta), \tag{8}$$

where $G_h^Q(s,a,\cdot)$ denotes the cumulant generating function of the random variable $Q_h^{M,\star}(s,a)$ (Kendall, 1946). Clearly K-learning and the 'RL as inference' framework are similar, since equations (6) and (7) are closedly linked, but there is a crucial difference. Notice that the integral performed in (8) is with respect to the *posterior* over $Q_h^{M,\star}(s,a)$, which includes the epistemic uncertainty explicitly.

Table 3: K-learning.

| | |
|---|---|
| Before episode $\ell$ | Calculate $\beta_\ell = \beta\sqrt{\ell}$ |
| Bellman equation | $K_h(s,a) = \mathbb{E}^\ell \mu(s,a) + \dfrac{\sigma^2\beta_\ell}{2n^\ell(s,a)} + \sum_{s'} \mathbb{E}^\ell \mathcal{P}(s',s,a)V^{\mathrm{K}}_{h+1}(s')$ |
| | $V^{\mathrm{K}}_h(s) = \beta_\ell^{-1} \log\sum_a \exp\beta_\ell K_h(s,a)$ |
| Policy | $\pi^{\mathrm{K}}_h(s,a) \propto \exp\beta_\ell K_h(s,a)$ |

Given the approximation to the posterior probability of optimality in (8) we could sample actions from it as our policy, as done by Thompson sampling (5). However, that requires computation of the cumulant generating function $G^Q_h(s,a,\beta)$, which is non-trivial. It was shown in (O'Donoghue, 2018) that an upper bound to the cumulant generating function could be computed by solving a particular 'soft' Bellman equation. The resulting K-values, denoted $K_h(s,a)$ at $(s,a,h)$, are also optimistic for the expected optimal Q-values. Specifically, for any sequence $\{\beta_\ell\}$ the following holds

$$K_h(s,a) \geq \beta_\ell^{-1} G^Q_h(s,a,\beta_\ell) \geq \mathbb{E}^\ell Q^{M,\star}_h(s,a). \tag{9}$$

Following a Boltzmann policy over these K-values satisfies a Bayesian regret bound which matches the current best bound for Thompson sampling up to logarithmic factors under the same set of assumptions. We summarize the K-learning algorithm in Table (3), where $\beta > 0$ is a constant and and $n^\ell(s,a)$ is the *visitation count* of $(s,a)$ before episode $\ell$, *i.e.*, the number of times the agent has taken action $a$ at state $s$, and $\sigma > 0$ is a constant. The uncertainty in the transition function is incorporated into the constant $\sigma$, which is a technical detail we omit here for clarity, see (O'Donoghue, 2018) for details. In this way the agent is given a reward signal that includes a bonus which is higher for states and actions that the agent has visited less frequently.

Comparing Tables 2 and 3 it is clear that soft Q-learning and K-learning share some similarities: They both solve a 'soft' value function and use Boltzmann policies. However, the differences are important. Firstly, K-learning has an explicit schedule for the inverse temperature parameter $\beta_\ell$, and secondly it adds a bonus based on visitation count to the expected reward. These two relatively small changes make K-learning a principled exploration and inference strategy.

To understand how K-learning drives exploration, consider its performance on Problem 1. Since this is a bandit problem we can compute the cumulant generating functions for each arm and then use the policy given by (8). For any non-trivial prior and choice of $\beta > 0$ the cumulant generating function is optimistic for arm 2 which results in the policy selecting arm 2 more frequently, thereby resolving its epistemic uncertainty. As $\beta \to \infty$ K-learning converges to the policy of pulling arm 2 deterministically. This is in contrast to soft Q-learning where arm 2 is exponentially *unlikely* to be selected as the exploration parameter $\beta$ grows.

### 3.3.1 CONNECTIONS BETWEEN K-LEARNING AND THOMPSON SAMPLING

Since K-learning can be viewed as approximating the posterior probability of optimality of each action it is natural to ask how close an approximation it is. A natural way to measure this similarity is the Kullback–Leibler (KL) divergence between the distributions,

$$D_{KL}(\mathbb{P}(\mathcal{O}_h(s)) \,||\, \pi^{\mathrm{K}}_h(s)) = \sum_a \mathbb{P}(\mathcal{O}_h(s,a))\log(\mathbb{P}(\mathcal{O}_h(s,a))/\pi^{\mathrm{K}}_h(s,a)),$$

where we are using the notation $\mathcal{O}_h(s) = \mathcal{O}_h(s,\cdot)$ and $\pi^{\mathrm{K}}_h(s) = \pi^{\mathrm{K}}_h(s,\cdot)$. This is different to the usual notion of distance that is taken in variational Bayesian methods, which would typically reverse the order of the arguments in the KL divergence (Blundell et al., 2015). However, in RL that 'direction' is not appropriate: a distribution minimizing $D_{KL}(\pi_h(s) \,||\, \mathbb{P}(\mathcal{O}_h(s)))$ may put zero probability on regions of support of $\mathbb{P}(\mathcal{O}_h(s))$. This means an action with non-zero probability of being optimal might *never* be taken. On the other hand a policy minimizing $D_{KL}(\mathbb{P}(\mathcal{O}_h(s)) \,||\, \pi_h(s))$ must assign a non-zero probability to every action that has a non-zero probability of being optimal, or incur an infinite KL divergence penalty. With this characterization in mind, and noting that the

Thompson sampling policy satisfies $\mathbb{E}^\ell \pi_h^{\mathrm{TS}}(s) = \mathbb{P}(\mathcal{O}_h(s))$, our next result links the policies of K-learning to Thompson sampling.

**Theorem 1.** *The K-learning value function $V^{\mathrm{K}}$ and policy $\pi^{\mathrm{K}}$ defined in Table 3 satisfy the following bound at every state $s \in \mathcal{S}$ and $h = 0, \dots H$:*

$$V_h^{\mathrm{K}}(s) \geq \mathbb{E}V_h^{M,\star}(s) + \beta^{-1} D_{KL}(\mathbb{P}(\mathcal{O}_h(s)) \,||\, \pi_h^{\mathrm{K}}(s)). \tag{10}$$

We defer the proof to Appendix 5.2. This theorem tells us that the distance between the true probability of optimality and the K-learning policy is bounded for any choice of $\beta < \infty$. In other words, if there is an action that might be optimal then K-learning will eventually take that action.

### 3.4 Why is 'RL as Inference' so popular?

The sections above outline some surprising ways that the 'RL as inference' framework can drive suboptimal behaviour in even simple domains. The question remains, why do so many popular and effective algorithms lie within this class? The first, and most important point, is that these algorithms can perform extremely well in domains where efficient exploration is not a bottleneck. Furthermore, they are often easy to implement and amenable to function approximation (Peters et al., 2010; Kober & Peters, 2009; Abdolmaleki et al., 2018). Our discussion of K-learning in Section 3.3 shows that a relatively simple fix to this problem formulation can result in a framing of RL as inference that maintains a coherent notion of optimality. Computational results show that, in tabular domains, K-learning can be competitive with, or even outperform Thompson sampling strategies, but extending these results to large-scale domains with generalization is an open question (O'Donoghue, 2018; Osband et al., 2017).

The other observation is that the 'RL as inference' can provide useful insights to the structure of particular *algorithms* for RL. It is valid to note that, under certain conditions, following policy gradient is equivalent to a dual inference problem where the 'probabilities' play the role of dummy variables, but are not supposed to represent the probability of optimality in the RL problem. In this light, Levine (2018) presents the inference framework as a way to generalize a wide range of state of the art RL algorithms. However, when taking this view, you should remember that this inference duality is limited to certain RL algorithms, and without some modifications (e.g. Section 3.3) this perspective is in danger of overlooking important aspects of the RL problem.

## 4 Computational experiments

### 4.1 One unknown action (Problem 1)

Consider the environment of Problem 1 with uniform prior $\phi = (\frac{1}{2}, \frac{1}{2})$. We fix $\epsilon = 1e - 3$ and consider how the Bayesian regret varies with $N > 3$. Figure 1 compares how the regret scales for Bayes-optimal (1.5), Thompson sampling (2.5), K-learning ($\leq 2.2$) and soft Q-learning (which grows linearly in $N$ for the optimal $\beta \to 0$, but would typically grow exponentially for $\beta > 0$). This highlights that, even in a simple problem, there can be great value in considering the value of information.

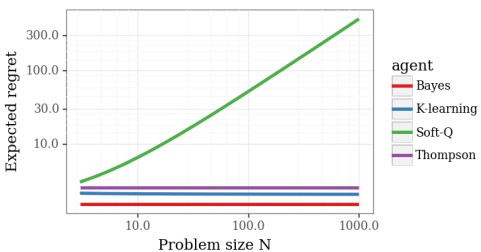

Figure 1: Regret scaling on Problem 1. Soft Q-learning does not scale gracefully with $N$.

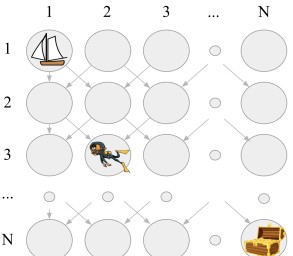

Figure 2: DeepSea exploration: A simple example where deep exploration is critical.

## 4.2 'DEEPSEA' EXPLORATION

Our next set of experiments considers the 'DeepSea' MDPs introduced by Osband et al. (2017). At a high level this problem represents a 'needle in a haystack', designed to require efficient exploration, the complexity of which grows with the problem size $N \in \mathbb{N}$. DeepSea (Figure 2) is a scalable variant of the 'chain MDPs' popular in exploration research (Jaksch et al., 2010). [6]

The agent begins each episode in the top-left state in an $N \times N$ grid. At each timestep the agent can move left or right one column, and falls one row. There is a small negative reward for heading right, and zero reward for left. There is only one rewarding state, at the bottom right cell. The only way the agent can receive positive reward is to choose to go right in each timestep. Algorithms that do not perform *deep exploration* will take an exponential number of episodes to learn the optimal policy, but those that prioritize informative states and actions can learn much faster.

Figure 3a shows the 'time to learn' for tabular implementations of K-learning (Section 3.3), soft Q-learning (Section 3.2) and Thompson sampling (Section 3.1). We implement each of the algorithms with a $N(0,1)$ prior for rewards and $\text{Dirichlet}(1/N)$ prior for transitions. Since these problems are small and tabular, we can use conjugate prior updates and exact MDP planning via value iteration. As expected, Thompson sampling and K-learning scale gracefully to large domains but soft Q-learning does not.

## 4.3 BEHAVIOUR SUITE FOR REINFORCEMENT LEARNING

So far our experiments have been confined to the tabular setting, but the main focus of 'RL as inference' is for scalable algorithms that work with generalization. In this section we show that the same insights we built in the tabular setting extend to the setting of deep RL. To do this we implement variants of Deep Q-Networks with a single layer, 50-unit MLP (Mnih et al., 2013). To adapt K-learning and Thompson sampling to this deep RL setting we use an ensemble of size 20 with randomized prior functions to approximate the posterior distribution over neural network Q-values (Osband et al., 2018) (full experimental details are included in Appendix 5.4). We then evaluate all of the algorithms on `bsuite`: A suite of benchmark tasks designed to highlight key issues in RL (Osband et al., 2019).

In particular, `bsuite` includes an evaluation on the DeepSea problems but with a one-hot pixel representation of the agent position. In Figure 3b we see that the results for these deep RL implementations closely match the observed scaling for the tabular setting. In particular, the algorithms motivated by Thompson sampling and K-learning both scale gracefully to large problem sizes, where soft Q-learning is unable to drive deep exploration. Our `bsuite` evaluation includes many more experiments that can be fit into this paper, but we provide a link to the complete results at `bit.ly/rl-inference-bsuite`. In general, the results for Thompson sampling and K-learning are similar, with soft Q-learning performing significantly worse on 'exploration' tasks. We push a summary of these results to Appendix 6.

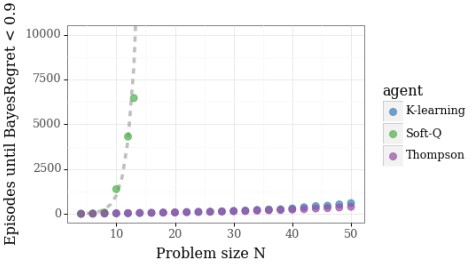
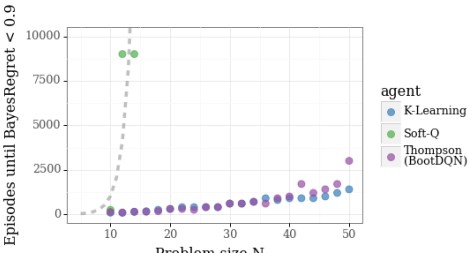

(a) Tabular state representation.      (b) One-hot pixel representation into neural net.

Figure 3: Learning times for DeepSea experiments. Dashed line represents $2^N$.

---

[6]DeepSea figure taken with permission from the 'bsuite' Osband et al. (2019)

## 5 CONCLUSION

This paper aims to make sense of reinforcement learning and probabilistic inference. We review the reinforcement learning problem and show that this problem of optimal learning already combined the problems of control and inference. As we highlight this connection, we also clarify some potentially confusing details in the popular 'RL as inference' framework. We show that, since this problem formulation ignores the role of epistemic uncertainty, that algorithms derived from that framework can perform poorly on even simple tasks. Importantly, we also offer a way forward, to reconcile the views of RL and inference in a way that maintains the best pieces of both. In particular, we show that a simple variant to the RL as inference framework (K-learning) can incorporate uncertainty estimates to drive efficient exploration. We support our claims with a series of simple didactic experiments. We leave the crucial questions of how to scale these insights up to large complex domains for future work.

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

## APPENDIX

### 5.1 SOFT Q-LEARNING DERIVATION

We present a derivation of soft Q-learning from the RL as inference parametric approximation to the probability of optimality. Although our presentation is slightly different to that of Levine (2018) we show here that the resulting algorithms are essentially identical. Recall from equation (6) that the parametric approximation to optimality we consider is given by

$$\tilde{\mathbb{P}}(\mathcal{O}_h(s,a)|\tau_h(s,a)) \propto \exp\left(\sum_{(s',a')\in\tau_h(s,a)} \beta\mathbb{E}^\ell\mu(s',a')\right)$$

$$= \exp(\beta\mathbb{E}^\ell\mu(s,a))\exp\left(\sum_{(s'',a'')\in\tau_{h+1}(s',a')} \beta\mathbb{E}^\ell\mu(s'',a'')\right)$$

$$= \exp(\beta\mathbb{E}^\ell\mu(s,a))\tilde{\mathbb{P}}(\mathcal{O}_{h+1}(s',a')|\tau_{h+1}(s',a'))$$

where $\tau_h(s,a)$ is a trajectory starting from $(s,a)$ at time $h$ and $\beta > 0$ is a hyper-parameter. Now we must marginalize out the possible trajectories $\tau_h$ using the (unknown) system dynamics. Since this is a certainty-equivalent algorithm we shall use the expected value of the transition probabilities, under the posterior at episode $\ell$, which means we can write

$$\tilde{\mathbb{P}}(\tau_h(s,a)) = \mathbb{E}^\ell\mathcal{P}(s',s,a)p(a'|s')\tilde{\mathbb{P}}(\tau_{h+1}(s',a')),$$

and we make the additional assumption that the 'prior' $p(a|s)$ is uniform across all actions $a$ for each $s$ (this assumption is standard in this framework, see Levine (2018)). In this case we obtain

$$\tilde{\mathbb{P}}(\mathcal{O}_h(s,a)) = \sum_{\tau_h(s,a)} \tilde{\mathbb{P}}(\mathcal{O}_h(s,a)|\tau_h(s,a))\tilde{\mathbb{P}}(\tau_h(s,a))$$

$$\propto \exp(\beta\mathbb{E}^\ell\mu(s,a))\sum_{s',a'}\mathbb{E}^\ell\mathcal{P}(s',s,a)\sum_{\tau'_{h+1}(s',a')}\tilde{\mathbb{P}}(\mathcal{O}_{h+1}(s',a')|\tau'_{h+1}(s',a'))\tilde{\mathbb{P}}(\tau'_{h+1}(s',a'))$$

$$= \exp(\beta\mathbb{E}^\ell\mu(s,a))\sum_{s',a'}\mathbb{E}^\ell\mathcal{P}(s',s,a)\tilde{\mathbb{P}}(\mathcal{O}_{h+1}(s',a')).$$

Now with this we can rewrite

$$\log\tilde{\mathbb{P}}(\mathcal{O}_h(s,a)) = \beta\mathbb{E}^\ell\mu(s,a) + \log\sum_{s',a'}\mathbb{E}^\ell\mathcal{P}(s',s,a)\tilde{\mathbb{P}}(\mathcal{O}_{h+1}(s',a')) - \log Z(s)$$

where $Z(s)$ is the normalization constant for state $s$, since $\sum_a\tilde{\mathbb{P}}(\mathcal{O}_h(s,a)) = 1$ for any $s$, and using Jensen's we have the following bound

$$\log\tilde{\mathbb{P}}(\mathcal{O}_h(s,a)) \geq \beta\mathbb{E}^\ell\mu(s,a) + \sum_{s'}\mathbb{E}^\ell\mathcal{P}(s',s,a)\log\sum_{a'}\tilde{\mathbb{P}}(\mathcal{O}_{h+1}(s',a')) - \log Z(s)$$

now if we introduce the soft Q-values that satisfy the soft Bellman equation

$$\tilde{Q}_h(s,a) = \mathbb{E}^\ell\mu(s,a) + \sum_{s'}\mathbb{E}^\ell\mathcal{P}(s',s,a)\beta^{-1}\log\sum_{a'}\exp\beta\tilde{Q}_{h+1}(s',a')$$

then

$$\tilde{\mathbb{P}}(\mathcal{O}_h(s,a)) \approx \exp\beta\tilde{Q}_h(s,a)/\sum_b\exp\beta\tilde{Q}_h(s,b)$$

and we have the soft Q-learning algorithm (the approximation comes from the fact we used Jensen's inequality to provide a bound).

### 5.2 PROOF OF THEOREM 1

**Theorem.** *The K-learning value function $V^K$ and policy $\pi^K$ defined in Table 3 satisfy the following bound at every state $s \in \mathcal{S}$ and $h = 0, \dots H$:*

$$V_h^K(s) \geq \mathbb{E}V_h^{M,\star}(s) + \beta^{-1}D_{KL}(\mathbb{P}(\mathcal{O}_h(s)) \,||\, \pi_h^K(s)).$$

*Proof.* Fix some particular state $s \in \mathcal{S}$, and let the joint posterior over value and optimality be denoted by

$$\mathbb{P}(V_h^{M,\star}(s), \mathcal{O}_h(s,a)) = \mathbb{P}(Q_h^{M,\star}(s,a)|\mathcal{O}_h(s,a))\mathbb{P}(\mathcal{O}_h(s,a)), \quad (11)$$

where we use $f$ to denote the conditional distribution over Q-values conditioned on optimality. Recall that from equation (7) we have approximated the conditional posterior probability of optimality as

$$\tilde{\mathbb{P}}(\mathcal{O}_h(s,a)|Q_h^{M,\star}(s,a)) \propto \exp \beta Q_h^{M,\star}(s,a),$$

for some $\beta > 0$, which when yields

$$\tilde{\mathbb{P}}(\mathcal{O}_h(s,a)) \propto \exp G_h^Q(s,a,\beta).$$

From Bayes' rule this implies the following approximation to the conditional distribution

$$\tilde{\mathbb{P}}(Q_h^{M,\star}(s,a)|\mathcal{O}_h(s,a)) = \frac{\tilde{\mathbb{P}}(\mathcal{O}_h(s,a)|Q_h^{M,\star}(s,a))\mathbb{P}(Q_h^{M,\star}(s,a))}{\tilde{\mathbb{P}}(\mathcal{O}_h(s,a))} \quad (12)$$
$$= \mathbb{P}(Q_h^{M,\star}(s,a))\exp(\beta Q_h^{M,\star}(s,a) - G_h^Q(s,a,\beta)).$$

This is known as the *exponential tilt* of the posterior distribution $\mathbb{P}(Q_h^{M,\star}(s,a))$ and has a myriad of applications in statistics (Asmussen & Glynn, 2007). From this we could derive an approximation to the joint posterior (11), however, the K-learning policy does not follow (8) since computing the cumulant generating function is non-trivial. Instead we compute the K-values, which are the solution to a Bellman equation that provide a guaranteed upper bound on the cumulant generating function, and the K-learning policy is thus

$$\pi_h^K(s,a) \propto \exp(\beta K_h(s,a)),$$

where we have (O'Donoghue, 2018)

$$\beta K_h(s,a) \geq G_h^Q(s,a)(\beta). \quad (13)$$

With that in mind we take our approximation to the joint posterior (11) to be

$$\tilde{\mathbb{P}}(V_h^{M,\star}(s), \mathcal{O}_h(s,a)) = \tilde{\mathbb{P}}(Q_h^{M,\star}(s,a)|\mathcal{O}_h(s,a))\pi_h^K(s,a).$$

Now consider the KL-divergence between the true joint posterior and our approximate one, a quick calculation yields

$$D_{KL}(\mathbb{P}(V_h^{M,\star}(s), \mathcal{O}_h(s,a)) \,||\, \tilde{\mathbb{P}}(V_h^{M,\star}(s), \mathcal{O}_h(s,a))) = D_{KL}(\mathbb{P}(\mathcal{O}_h(s)) \,||\, \pi_h^K(s)) +$$
$$\sum_a \mathbb{P}(\mathcal{O}_h(s,a))D_{KL}(\mathbb{P}(Q_h^{M,\star}(s,a)|\mathcal{O}_h(s,a)) \,||\, \tilde{\mathbb{P}}(Q_h^{M,\star}(s,a)|\mathcal{O}_h(s,a))), \quad (14)$$

for timestep $h$ and state $s$. Considering the terms on the right hand side of (14) separately we have

$$D_{KL}(\mathbb{P}(\mathcal{O}_h(s)) \,||\, \pi_h^K(s)) = -\mathcal{H}(\mathbb{P}(\mathcal{O}_h(s))) - \beta \sum_a \mathbb{P}(\mathcal{O}_h(s,a))K_h(s,a) + \log \sum_a \exp \beta K_h(s,a)$$

where $\mathcal{H}$ denotes the entropy, and using (12)

$$\sum_a \mathbb{P}(\mathcal{O}_h(s,a))D_{KL}(\mathbb{P}(Q_h^{M,\star}(s,a)|\mathcal{O}_h(s,a)) \,||\, \tilde{\mathbb{P}}(Q_h^{M,\star}(s,a)|\mathcal{O}_h(s,a)))$$
$$= \sum_a \mathbb{P}(\mathcal{O}_h(s,a))G_h^Q(s,a)(\beta) - \beta \sum_a \mathbb{P}(\mathcal{O}_h(s,a))\mathbb{E}(Q_h^{M,\star}(s,a)|\mathcal{O}_h(s,a))$$
$$+ \sum_a \mathbb{P}(\mathcal{O}_h(s,a))D_{KL}(\mathbb{P}(Q_h^{M,\star}(s,a)|\mathcal{O}_h(s,a)) \,||\, \mathbb{P}(Q_h^{M,\star}(s,a))).$$

Now we sum these two terms, using (13) and the following identities

$$\sum_a \mathbb{P}(\mathcal{O}_h(s,a))\mathbb{E}(Q_h^{M,\star}(s,a)|\mathcal{O}_h(s,a)) = \mathbb{E}\max_a Q_h^{M,\star}(s,a) = \mathbb{E}V_h^{M,\star}(s)$$

and

$$\sum_a \mathbb{P}(\mathcal{O}_h(s,a)) D_{KL}(\mathbb{P}(Q_h^{M,\star}(s,a)|\mathcal{O}_h(s,a)) \,||\, \mathbb{P}(Q_h^{M,\star}(s,a)))$$

$$= \sum_a \mathbb{P}(\mathcal{O}_h(s,a)) \int \mathbb{P}(Q_h^{M,\star}(s,a)|\mathcal{O}_h(s,a)) \log(\mathbb{P}(\mathcal{O}_h(s,a)|Q_h^{M,\star}(s,a))) + \mathcal{H}(\mathbb{P}(\mathcal{O}_h(s)))$$

$$\leq \mathcal{H}(\mathbb{P}(\mathcal{O}_h(s))),$$

since $\log(\mathbb{P}(\mathcal{O}_h(s,a)|Q_h^{M,\star}(s,a))) \leq 0$, we obtain

$$D_{KL}(\mathbb{P}(V_h^{M,\star}(s), \mathcal{O}_h(s,a)) \,||\, \tilde{\mathbb{P}}(V_h^{M,\star}(s), \mathcal{O}_h(s,a))) \leq \log \sum_a \exp \beta K_h(s,a) - \beta \mathbb{E} V_h^{M,\star}(s).$$

The theorem follows from this and the fact that the K-learning value function is defined as

$$V_h^{\mathrm{K}}(s) = \beta^{-1} \log \sum_a \exp \beta K_h(s,a)$$

as well as the fact that

$$D_{KL}(\mathbb{P}(\mathcal{O}_h(s)) \,||\, \pi_h^{\mathrm{K}}(s)) \leq D_{KL}(\mathbb{P}(V_h^{M,\star}(s), \mathcal{O}_h(s,a)) \,||\, \tilde{\mathbb{P}}(V_h^{M,\star}(s), \mathcal{O}_h(s,a)))$$

from equation (14). $\qquad\qquad\square$

## 5.3 PROBLEM 1 K-LEARNING DETAILS

For a bandit problem the K-learning policy is given by

$$\pi_i^{\mathrm{K}} \propto \exp G_i^{\mu}(\beta),$$

which requires the cumulant generating function of the posterior over each arm. For arm 1 and the distractor arms there is no uncertainty, in which case the cumulant generating function is given by

$$G_i^{\mu}(\beta) = \mu_i \beta, \quad i = 1, 3, \dots N.$$

In the case of arm 2 the cumulant generating function is

$$G_2^{\mu}(\beta) = \log \big((1/2) \exp(2\beta) + (1/2) \exp(-2\beta)\big).$$

In (O'Donoghue, 2018) it was shown that the optimal choice of $\beta$ is given by

$$\beta^{\star} = \operatorname*{argmin}_{\beta \geq 0} \left( \beta^{-1} \log \sum_{i=1}^{N} \exp G_i^{\mu}(\beta) \right),$$

which requires solving a convex optimization problem in variable $\beta^{-1}$. In the case of problem 1 the optimal choice of $\beta \approx 10.23$, which yields $\pi_2^{kl} \approx 0.94$. Then, once arm 2 has been pulled once and the true reward of arm 2 has been revealed, its cumulant generating function has the same form as the others, and then the optimal choice of $\beta$ is simply

$$\beta^{\star} = \operatorname*{argmin}_{\beta \geq 0} \left( \beta^{-1} \log \sum_{i=1}^{N} \exp \mu_i \beta \right) = \infty,$$

at which point K-learning is greedy with respect to the optimal arm.

## 5.4 IMPLEMENTATION DETAILS FOR BSUITE EVALUATION

All three algorithms use the same neural network architecture consisting of an MLP (multilayer perceptron) with a single hidden layer with 50 hidden units. All three algorithms used a replay buffer of the most recent $10^4$ transitions to allow re-use of data. For all three the Adam optimizer (Kingma & Ba, 2014) was used with learning rate $10^{-3}$ and batch-size 128, and learning is performed at every time-step. For both K-learning and soft Q-learning the temperature was set at $\beta^{-1} = 0.01$. For Bootstrap DQN we chose an ensemble of size 20, and used the randomized prior functions (Osband et al., 2018) with scale 3.. For K-learning, in order to estimate the cumulant generating function of the reward, we used an ensemble of neural networks predicting the reward for each state and action and used these to calculate the empirical cumulant generating function over them. Each of these was a single hidden layer MLP with 10 hidden units. Finally, we noted that actually training a small ensemble of K-networks performed better than a single network, we used an ensemble of size 10 for this purpose as well as using randomized priors to encourage diversity between the elements of the ensemble with scale 1.0. The K-learning policy was the Boltzmann policy over all the ensemble K-values at each state.

# 6 `bsuite` report: Making sense of RL and Inference

The *Behaviour Suite for Reinforcement Learning*, or `bsuite` for short, is a collection of carefully-designed experiments that investigate core capabilities of a reinforcement learning (RL) agent. The aim of the `bsuite` project is to collect clear, informative and scalable problems that capture key issues in the design of efficient and general learning algorithms and study agent behaviour through their performance on these shared benchmarks. This report provides a snapshot of agent performance on `bsuite2019`, obtained by running the experiments from `github.com/deepmind/bsuite` Osband et al. (2019).

## 6.1 AGENT DEFINITION

All agents were run with the same network architecture (a single layer MLP with 50 hidden units a ReLU activation) adapting DQN (Mnih et al., 2013). Full hyperparameters in Appendix 5.4.

- **boot_dqn**: bootstrapped DQN with prior networks (Osband et al., 2016; 2018).
- **k_learn**: K-learning via ensemble with prior networks (O'Donoghue, 2018; Osband et al., 2018).
- **soft_q**: soft Q-learning with temperature $\beta^{-1} = 0.01$ (O'Donoghue et al., 2017).

## 6.2 SUMMARY SCORES

Each `bsuite` experiment outputs a summary score in [0,1]. We aggregate these scores by according to key experiment type, according to the standard analysis notebook. A detailed analysis of each of these experiments may be found in a notebook hosted on Colaboratory: `bit.ly/rl-inference-bsuite`.

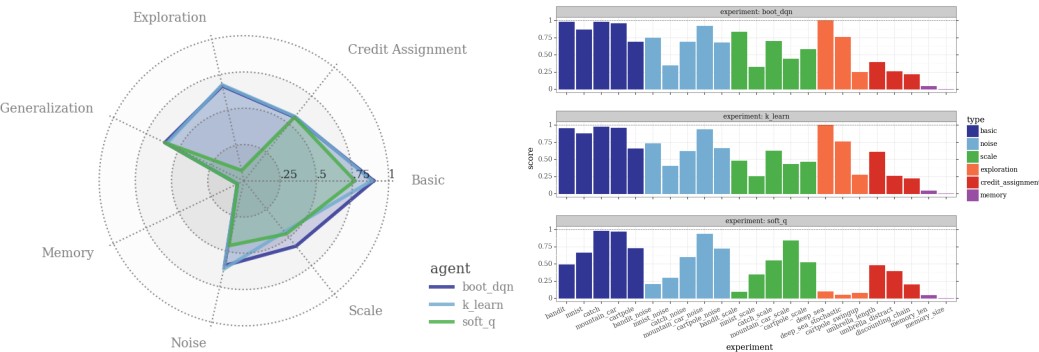

Figure 4: Snapshot of agent behaviour.   Figure 5: Score for each `bsuite` experiment.

## 6.3 RESULTS COMMENTARY

Overall, we see that the algorithms K-learning and Bootstrapped DQN perform extremely similarly across `bsuite` evaluations. However, there is a clear signal that soft Q-learning performs markedly worse on the tasks requiring efficient exploration. This observation is consistent with the hypothesis that algorithms motivated by 'RL as Inference' fail to account for the value of exploratory actions.

Beyond this major difference in exploration score, we see that Bootstrapped DQN outperforms the other algorithms on problems varying 'Scale'. This too is not surprising, since both soft Q and K-learning rely on a temperature tuning that will be problem-scale dependent. Finally, we note that soft Q also performs worse on some 'basic' tasks, notably 'bandit' and 'mnist'. We believe that the relatively high temperature (tuned for best performance on Deep Sea) leads to poor performance on these tasks with larger action spaces, due to too many random actions.

