# OpenReview forum: "Making Sense of Reinforcement Learning and Probabilistic Inference"
_ICLR.cc/2020/Conference — Accept (Spotlight)_

### Official Review · AnonReviewer1 · 2019-10-17
**Official Blind Review #1**

**Rating:** 8

**Review:**

This paper criticizes the ‘RL as inference’ paradigm by highlighting its limitations and shows that a variant of this framework - the K-learning algorithm (O'Donoghue et. al., 2018) does not have these limitations. The paper first clarifies some points of confusion regarding RL as inference, namely the fact that RL was originally an inference problem all along. A simple example is used to demonstrate that the RL as inference framework (Levine, 2018) fails to choose the optimal actions that resolve epistemic uncertainty, whereas the K-learning algorithm does select the optimal action. Further, a connection is made which reveals that K-learning is an approximate version of Thompson sampling - the strategy of using as single posterior sample of parameters given data for greedy actions which originated in bandit settings. Some empirical results are provided highlighting the cases where Soft Q-learning (Levine, 2018) fails but Thompson sampling and K-learning do not.

I vote for accepting this paper as it brings to light an important limitation of the popular RL as Inference framework with a didactic example which, to the best of my knowledge, has not been shown before.

The paper does a great job at succinctly introducing a simple bandit problem where the bayes-optimal policy is to take a first action that is supposed to immediately resolve all epistemic uncertainty and then exploit the optimal action repeatedly for future plays. However, this simple problem is designed in such a way that there are several other sub-optimal actions which make the RL as inference algorithm have an exponentially low probability of selecting the optimal action. This implies that RL as inference, unlike Thompson sampling, does not in fact take into account epistemic uncertainty.

Feedback to authors:
- The introduction of family of MDPs caused a lot of confusion about the problem setting. I was not sure if a new MDP is sampled from \phi at every episode in L or a single MDP is sampled and kept the same throughout. This was clarified later on in the middle of section 2.1, but it could have been introduced more carefully earlier on,
- The tables 1-3 summarizing algorithms are useful but it would be great if there could be a side by side comparison of all three in a single table.
- The notation is very dense and I see that efforts were made to avoid this, but it still feels inaccessible.
- I am not sure of the role of experiments in section 4.3, if there is no comparison to K-learning. I understand that the authors leave it to future work but then the experiments feel out of place.
- “... RL as inference has inspired many interesting and novel techniques, as well as delivered algorithms with good performance on problems where exploration is not the bottleneck (Gregor et al., 2016)”. I think this sentence is false, Gregor et. al. do not employ RL as inference anywhere in their paper. Also, I don’t think the point of their paper was to show good performance on any problem. Maybe this was mixed up with Eysenbach, 2018, a successor paper which uses RL as inference?



References:
O'Donoghue, Brendan. "Variational Bayesian Reinforcement Learning with Regret Bounds." arXiv preprint arXiv:1807.09647 (2018).

Levine, Sergey. "Reinforcement learning and control as probabilistic inference: Tutorial and review." arXiv preprint arXiv:1805.00909 (2018).

Gregor, Karol, Danilo Jimenez Rezende, and Daan Wierstra. "Variational intrinsic control." arXiv preprint arXiv:1611.07507 (2016).

Eysenbach, Benjamin, et al. "Diversity is all you need: Learning skills without a reward function." arXiv preprint arXiv:1802.06070 (2018).

**Experience Assessment:**

I have read many papers in this area.

**Review Assessment: Checking Correctness Of Derivations And Theory:**

I assessed the sensibility of the derivations and theory.

**Review Assessment: Checking Correctness Of Experiments:**

I assessed the sensibility of the experiments.

**Review Assessment: Thoroughness In Paper Reading:**

I read the paper thoroughly.

---

> ### Author Response · Authors · 2019-11-09
> **Re: Official Blind Review #1**
>
> Thank you for your detailed review, and for your kind words! In response to your concerns:
>
> You are right, this was confusing. Indeed the *true* MDP is sampled only once and is the same for all learning thereafter, though Thompson sampling samples an MDP from the posterior at each iteration as part of the learning. We have clarified this earlier in the paper.
>
> We have tried to merge all three algorithms into a single table, but it’s quite dense. We’ll keep experimenting with it and hopefully we’ll have a good compromise by the time we resubmit a revision, which we’re hoping to upload in the next couple of days, though it might just be the case that three separate tables is cleanest unfortunately!
>
> We have made efforts to clean up the notation, especially with respect to the derivation of soft Q-learning.
> We have added a form of deep K-learning and Bootstrapped DQN to the results of section 4.3 to suggest that our claims likely carry over to the function approximation case.
>
> You are right, that’s not the correct reference at that point, we have updated it to the Eysenbach 2018 paper.

---

### Official Review · AnonReviewer2 · 2019-10-23
**Official Blind Review #2**

**Rating:** 6

**Review:**

The paper at hand presents an alternative view on reinforcement learning as probabilistic inference (or equivalently maximum entropy reinforcement learning). With respect to other formulations of this view (e.g. Levine, 2018; I am referring to the references of the paper here), the paper identifies a shortcoming in the disregard of the agent’s epistemic uncertainty (which seems to refer to the uncertainty with respect to the underlying MDP). It is argued, that algorithms based on the prevailing probabilistic formulation (e.g. soft Q-learning) suffer from suboptimal exploration.
The paper thus compares maximum entropy RL to K-learning (O’Donoghue, 2018), which is taken to address the issue of suboptimal exploration due to its temperature scheduling and its inclusion of state-action pair counts in the reward signal.

As its technical contribution, the paper re-interprets K-learning via the latent variable denoting optimality employed in Levine (2018) and introduces a theorem bounding the distance between the policies of Thompson sampling and K-learning. Empirical validation of the claims is provided via experiments on an engineered bandit problem and the tabular MDP (i.e. DeepSea from Osband et al., 2017), as well as via soft Q-learning results on the recently suggested bsuite (Osband et al., 2019).

I consider this paper a weak reject. This is in light of me finding it very hard to follow the papers main claims and arguments, even though it positions itself as communicating connections (“making sense”) in prior work, rather than presenting a novel algorithm. While this is in part due to the complicated issue and math being discussed (and the paper probably catering to a very narrow audience), the paper in its current state does seem to hinder understanding as well.

On the positive side, I do appreciate the intention of the paper, namely to connect RL as probabilistic inference, Thompson sampling and K-learning. In my opinion, this can be taken as a valuable addition to the current understanding of these approaches. Also, I like the experiments as they are specifically constructed to support the claims of the paper.
On the negative side, vague language, missing assumptions and lax notation seem to hinder the understanding of the paper to a considerable extend: e.g. it is stated, that “we connect the resulting algorithm with […] K-learning”. However, I do not recognize a new algorithm being provided. Instead the paper argues in favor of K-learning. The assumptions that come with K-learning are not mentioned. The restriction of K-learning to tabular RL is taken to be understood implicitly (whereas RL as probabilistic inference seems applicable with function approximation also, which is not mentioned in the comparison). The paper always talks of shortcomings (plural) of RL as probabilistic inference, but only provides one argument (suboptimal exploration) with respect to this. RL as probabilistic inference is introduced in a different form as in prior literature (i.e. Equation 6), while the derivation in the Appendix spanning the differences in notation being hard to follow due to (maybe minor?) notational issues (e.g. x and y seem to have replaced s’ and a; further down there is a reference to Equation 7, however probably it is meant to be 8 and even that with some leap in notation).
The paper would benefit from better proof-reading, where mistakes in a very dense argumentation make it hard to follow (e.g. I do not understand the sentence “The K-learning expectation (7) is with respect to the posterior over Q[…] to give a parametric approximation to the probability of optimality.”)

Literature wise, the paper draws heavily from two unpublished papers (Levine, 2018; O’Donoghue, 2018). While this makes it harder to arrive at a high confidence level with respect to the paper’s claims, I would not argue this to be critical.
I would consider raising my score, if the authors would improve the accessibility of the paper by polishing the argumentation and notation.

Confidence: low. It is very likely, that I have misunderstood key arguments and derivations. Also, I did not attempt to follow all of the technical derivations.


======
post rebuttal comment:


I changed the score of my review in light of the rebuttal.
The changes made to the paper overall address my concerns.
I do consider the additional explanations and re-phrasings as well as the improved notation a nice improvement of the paper.
While I did not read all of the appendix, Section 5.1 is much more readable and understandable in the new version.

In light of this paper probably being published, I share some typos/inconsistencies I still noticed:

p. 4: the solving for -> solving for the
p. 7: s_{h+1} -> s' (in Table 3) ?
p. 7: table -> Table; tables -> Tables
p. 7: (Fix position of K:) \pi_h(s)^K -> \pi_h^K(s) ((also in Appendix))
p. 9: (2x) soft-Q learning -> soft Q-learning; Q Networks -> Q-Networks; Soft Q -> soft Q-learning




**Experience Assessment:**

I have read many papers in this area.

**Review Assessment: Checking Correctness Of Derivations And Theory:**

I assessed the sensibility of the derivations and theory.

**Review Assessment: Checking Correctness Of Experiments:**

I assessed the sensibility of the experiments.

**Review Assessment: Thoroughness In Paper Reading:**

I read the paper at least twice and used my best judgement in assessing the paper.

---

> ### Author Response · Authors · 2019-11-09
> **Re: Official Blind Review #2**
>
> Thank you for your careful and thorough review. We have tried to address your major concern, which was the lack of accessibility of the paper, by
>
> Making the assumptions clearer
> Fixing the notation
> Improving the clarity of the language overall.
>
> We have also tried to make it clear that we are not proposing a new algorithm, merely highlighting a shortcoming about RL as inference, and contrasting with alternative approaches. Moreover, it is not our intention to claim that K-learning is by any means the solution to the exploration problem, just that it, along with Thompson sampling (and other algorithms that we don’t focus on) do actually explore in a directed manner. We also make the connection of K-learning to the current ‘RL as inference’ framework as it is currently understood in the literature. The reason we do this is primarily because the RL as inference framework has inspired many new and interesting algorithms in many subfields of RL, including hierarchical RL, options / skills, multi-agent, empowerment etc. We hope that highlighting a major issue that the framework suffers from, and demonstrating that a fix is possible, that the performance of these algorithms can be improved further with (hopefully) relatively little change.
>
> As for the tabular vs function approximation issue, we do have a section entitled ‘Why is RL as inference so popular?’, in which we say ‘Further [the RL as inference derived algorithms] are often easy to implement and amenable to function approximation’. We want to stress that we understand that the RL as inference framework actually has a lot of value, but that the issue of sub-optimal exploration needs to be addressed. It is an open question as to how best to implement Thompson sampling and K-learning with function approximation - we have made this point more clear, though in response to your concern we have added some experiments with function approximation for both K-learning and Thompson sampling. These experiments suggest that it is at least possible.
>
> You are correct that we have slightly modified the presentation of RL as inference. This is to make it easier to compare with the other techniques we compare against. However, the formulation is not new. The paper by Deisenroth et al. 2013 “A survey on policy search for robotics” uses the same presentation. In section 2.4.2.2 the authors of that paper propose P(O = 1| tau) \propto exp(R(tau)), where R(tau) is the reward along the entire trajectory tau (they use an overloaded R instead of O to denote ‘optimality’, but other than that it is identical). The point here is that although the presentations differ ultimately the framework is the same. We have updated the comment to be clearer on this, and fixed the derivation of soft Q-learning in the appendix.

---

### Official Review · AnonReviewer3 · 2019-10-29
**Official Blind Review #3**

**Rating:** 6

**Review:**

The authors develop a criticism of the "RL as inference" standard approximations and propose a simple modification that solves its main issues while keeping hold of its advantage. Even though this modification ends up relating to a previously published algorithm, I judge the submission to be worthwhile publishing for the following contributions:
- clarity/didacticism of the exposition, the minimal problem, the positioning,
- the theorem,
- the (hopefully to be completed) experiments

The experiments are my main criticism of the paper, in particular the bsuite ones that was absolutely impenetrable for me: not only the experiments but also the results. I hope this will be completed in the final version. It was also a bit unclear to me the advantage of K-learning over Thomson sampling methods.

Minor remarks and typos:
- famliy => family
- I would not say that frequentist RL is the worst-case, but more high-probability (it's the worst case within the concentration bounds).
- the agent in then => the agent is then
- KL has 2 meanings in the notations: K-learning and KL divergence. For clarity, I suggest to use only K for K-learning (for instance).

**Experience Assessment:**

I have read many papers in this area.

**Review Assessment: Checking Correctness Of Derivations And Theory:**

I assessed the sensibility of the derivations and theory.

**Review Assessment: Checking Correctness Of Experiments:**

I did not assess the experiments.

**Review Assessment: Thoroughness In Paper Reading:**

I read the paper at least twice and used my best judgement in assessing the paper.

---

> ### Author Response · Authors · 2019-11-09
> **Re: Official Blind Review #3**
>
> Thank you for your review and for your suggestions. We completely agree that the experiments needed to be improved, in particular the bsuite ones. We have put a significant amount of work into improving them, and making it clear what the point of each one was, as well as presenting the results of the experiments more clearly. Moreover, we have added new experiments using function approximation for both K-learning and Thompson sampling which suggest that the claims we make in the paper in the tabular case carry over to the function approximation case, at least empirically.
>
> We do not make any claims about the advantages of K-learning or Thompson sampling over each other, they appear to have similar performance empirically, and the best RL Bayesian regret bounds for each algorithm are identical (up to logarithmic terms). We have tried to be clearer about this in the manuscript. The reason we mention them is to highlight the differences (and similarities where appropriate) between these two techniques and the RL as inference framework, which is unable to handle epistemic (Bayesian) uncertainty.
>
> Thank you also for the minor remarks and your eagle eye in spotting typos, we have corrected or clarified all of them!

---

### Author Response · Authors · 2019-11-09
**Thank you to the reviewers**

We would like to thank the reviewers for their careful consideration of our paper. All three reviewers highlighted important points that, once addressed, will improve the quality of our manuscript. We would also like to thank the reviewers for their kind words on the value of the paper. In particular both R1 and R3 both highlight the overall clarity of exposition and how the minimal examples highlight the issues we want to address, and R2 noted that the paper will be a valuable addition to the current understanding of RL and inference.

We have responded below to each reviewer and we have outlined the changes to the paper we will make in response. We are hoping to upload a new revision in the next couple of days. Overall, we are putting in a significant effort to improve clarity, notation, and accessibility. Furthermore, we have significantly improved the experiments, which we agree were confusing. In particular, we have added a deep implementation of K-learning and a deep variant of Thompson sampling (both using neural net function approximation) to the experimental results. The take home message is that both these implementations improve over soft-Q-learning when it comes to exploration, even when using function approximation.

---

> ### Author Response · Authors · 2019-11-12
> **Uploaded new revision**
>
> Thanks again to the reviewers, we have just uploaded a new revision which addresses most of the reviewers concerns.

---

### Public Comment · ~Matthew_Fellows1 · 2019-12-21
**Important Related Work**

In our recent NeurIPS paper, VIREL: A Variational Inference Framework for Reinforcement Learning (Fellows et al. 2019), we provide a discussion of the shortcomings of the maximum entropy reinforcement learning (MERL) framework as well as a simple counterexample to demonstrate that for several MDPs, the optimal policy under the classical RL objective cannot be recovered from the optimal MERL policy. We also introduce an alternative framework that does not suffer from the issues of existing RL and inference methods. This seems very relevant to the discussion in Section 3.3 in this work.

---

> ### Author Response · Authors · 2019-12-21
> **Thank you for the reference**
>
> I do not think we were aware of your paper, but we will make sure to investigate this!
>
> Do you have a summary for how you feel this work is related? Maybe we can have this discussion directly via email?
>
> Many thank!

---

### Decision · Program_Chairs · 2019-12-19

**Decision:**

Accept (Spotlight)

**Comment:**

The paper explores in more detail the "RL as inference" viewpoint and highlights some issues with this approach, as well as ways to address these issues. The new version of the paper has effectively addressed some of the reviewers' initial concerns, resulting in an overall well-written paper with interesting insights.